# OpenReview forum: "Personalized Safety in LLMs: A Benchmark and A Planning-Based Agent Approach"
_NeurIPS.cc/2025/Conference — NeurIPS 2025 poster_

### Official Review · Reviewer_hPDn · 2025-06-25

**Clarity:** 2
**Significance:** 3
**Originality:** 4
**Rating:** 4
**Confidence:** 2

**Summary:**

*I was added as a reviewer later in the process and did not part-take in the bidding process. I'm not familiar with the research area of this submission (safety and personalization).*

This paper introduces the concept of "personalized safety" for large language models (LLMs), addressing the critical issue that identical responses can have divergent risks depending on the user's background or condition. The authors present two main contributions:

1. PENGUIN Benchmark: A comprehensive dataset of 14,000 scenarios across seven high-stakes domains (e.g., health, finance, relationships), each with context-rich and context-free variants. This benchmark enables systematic evaluation of LLM safety in personalized settings, capturing dimensions like risk sensitivity, emotional empathy, and user-specific alignment.
2. RAISE Framework: A training-free, two-stage agent framework designed to improve personalized safety. RAISE uses Monte Carlo Tree Search (MCTS) to strategically acquire user-specific context attributes, optimizing for safety under limited interaction budgets. The framework dynamically determines when sufficient context is gathered to generate safe responses, achieving improvements in safety scores over vanilla LLMs while maintaining low interaction costs (2.7 queries on average).

The paper demonstrates that access to user context significantly enhances safety and highlights the need for selective information gathering, as not all attributes contribute equally to safety. The work establishes a foundation for adapting LLM safety mechanisms to individual user contexts.

**Questions:**

None

**Ethical Concerns:**

["NO or VERY MINOR ethics concerns only"]

**Final Justification:**

This paper introduces Personalized Safety as a new and important problem, proposing both a benchmark and a method. While the authors have made considerable efforts, uncertainties remain regarding: (1) whether the benchmark adequately reflects real-world distributions, and (2) the fundamental soundness of the proposed method. Therefore, I give a BA score.

**Limitations:**

yes

**Quality:**

3

**Strengths And Weaknesses:**

Strengths

* The paper introduces personalized safety as a new and important task for LLMs, motivated by real-world risks where generic responses can harm vulnerable users (e.g., mental health crises).

- They build a new benchmark (PENGUIN) for this new task, covering seven high-stakes domains (health, finance, relationships, etc.), with both real-world (Reddit) and synthetic data. It also includes both context-rich and context-free setup for controlled evaluation.
- Some valuable insights are revealed. The analysis shows that user context can improves safety, but not all attributes contribute equally.

Weaknesses

- The benchmark includes synthetic scenarios generated by an LLM, which may inherit biases or unrealistic patterns from the model.
- The explanation of RAISE is hard to follow.

- RAISE assumes users will iteratively provide personal details, which may be unrealistic in practice due to privacy concerns or reluctance. If users are willing to disclose sensitive information, a simple form might be more efficient than a multi-turn LLM interaction.

---

> ### Author Rebuttal · Authors · 2025-07-30
>
> We thank the reviewer for the insightful feedback and thorough review for our paper. We address the reviewer’s concerns and questions below:
>
> >**The benchmark includes synthetic scenarios generated by an LLM, which may inherit biases or unrealistic patterns from the model.**
>
> Thank you for raising this concern. To mitigate the risk that LLM-generated scenarios may inherit biases or unrealistic patterns, **synthetic cases constitute only 50% of the benchmark**. All generated data is produced under **relational constraints** to ensure **logical consistency across user attributes** (e.g., age–education alignment, emotional state–mental health coherence) (**Appendix A.3.1**). In addition, we applied manual filtering and diversity checks to eliminate implausible outputs. Combined with **real Reddit data from 16 communities** (**Appendix A.2**), these measures help maintain both realism and broad coverage.
>
> >**The explanation of RAISE is hard to follow.**
>
> Thank you for this feedback. To make RAISE easier to follow, we will provide a concise summary of its purpose—a two-stage framework that plans which user attributes to collect under strict query budgets and uses an abstention mechanism to decide when to stop. Besides, we will move essential content from **Appendix C and D** into the main text. This includes the workflow diagram, a simplified version of the LLM-Guided MCTS pseudocode (Appendix C.4), and details on the abstention logic and attribute selection process (Appendix D.1/D.2). These changes will make the interaction loop between modules clear while keeping the technical content intact.
>
> >**RAISE assumes users will iteratively provide personal details, which may be unrealistic in practice due to privacy concerns or reluctance. If users are willing to disclose sensitive information, a simple form might be more efficient than a multi-turn LLM interaction.**
>
> Thank you for raising this point. Unlike **static forms**, which typically request **all fields upfront**—often including **irrelevant or highly sensitive information**—RAISE is designed to minimize user disclosure. It adaptively selects only the most essential attributes under strict query budgets and stops as soon as sufficient context for safety is reached. **Prior studies** show that large, static forms increase cognitive load and lead to privacy fatigue and higher refusal rates [1, 2]. By contrast, RAISE’s incremental strategy ensures that safety-critical details (e.g., mental health indicators) are only requested when absolutely necessary, providing a better privacy–usability trade-off.
>
> [1] Hanbyul Choi, et al. (2018). The role of privacy fatigue in online privacy behavior. Computers in Human Behavior. Volume 81, Pages 42-51.
>
> [2] Jie Tang, et al. (2021). Why people need privacy? The role of privacy fatigue in app users' intention to disclose privacy: based on personality traits. Journal of Enterprise Information Management 34 (4): 1097–1120.
> .

---

> > ### Comment · Reviewer_hPDn · 2025-08-04
> >
> > Thank you for your detailed clarifications. Below are my follow-up questions:
> > * You mentioned several efforts, including "relational constraints to ensure logical consistency across user attributes" and "manual filtering and diversity checks." But how can you guarantee that the data, after these steps, truly reflects real-world distributions?
> >
> > * Why do you assume that forms must be static and include all fields? If a dynamic, refined form were designed, wouldn’t it be more convenient than RAISE? Furthermore, all of this relies on the assumption that users are willing to repeatedly provide personal information. What do you think of this premise?

---

> ### Author Response · Authors · 2025-08-04
> **Replying to follow-up questions**
>
> >**Concern with real-world distributions**
>
> Thank you for raising this important question. Our goal is **not to replicate overall real-world frequencies**, but to **stress-test models under rare yet critical high-risk conditions**. Prior work [1, 2] shows that safety failures are **low-frequency but high-impact**, so matching real-world distributions would overrepresent low-risk queries and underrepresent the scenarios most relevant to safety evaluation.
>
> To maintain realism within this risk-focused space, we combine real Reddit cases (50%; Appendix A.2) with synthetic cases generated under relational constraints and diversity checks (Appendix A.3) to ensure logical consistency (e.g., age–education alignment) and scenario diversity. This design emphasizes broad coverage of diverse risk profiles to prevent blind spots.
>
> Importantly, the observed improvements are not an artifact of synthetic data: performance trends remain consistent when evaluated on the Reddit-only subset, supporting the robustness of our approach.
>
> >**Why do you assume that forms must be static and include all fields? If a dynamic, refined form were designed, wouldn’t it be more convenient than RAISE?**
>
> Thank you for this suggestion. While dynamic forms are a reasonable direction for improving user experience, two key concerns remain:
>
> **If a dynamic form attempts to cover all possible cases, it faces significant scalability challenges**. As more attributes and risk factors are introduced, the number of possible combinations grows **exponentially** (e.g., even current 10 attributes with multiple states produce millions of paths), making rule-based branching impractical. Even coarse grouping leads to degraded performance, as shown by our fixed-path baselines in Figure 6 (Section 5.4), which perform significantly worse than fine-grained adaptive strategies.
>
> **If these paths are not predefined, real-time adaptation is required** because each subsequent question depends on previously collected user information to minimize unnecessary queries. For example, if a user reports high stress early in emotion, the system should prioritize mental health attributes immediately, whereas a stable response shifts focus to other dimensions. Fixed or rule-based logic cannot support such personalized, efficiency-driven planning, while RAISE achieves this adaptively through LLM-guided MCTS and an Abstention Module (Section 5, Figure 7). Empirically, RAISE reduces acquisition steps by an average of 2.7 queries (from a 10-step budget) without degrading safety performance (Section 5.4).
>
> Importantly, RAISE remains extensible as new attributes are added, whereas rule-based forms require extensive manual redesign. **A dynamic UI could still leverage RAISE as its backend logic to design the real-time adaptation dynamic forms**, which we will clarify in the revision.
>
> >**Furthermore, all of this relies on the assumption that users are willing to repeatedly provide personal information. What do you think of this premise?**
>
> Thank you for raising this concern. We **do not assume that users are willing to disclose unlimited personal information**. In fact, **RAISE is explicitly designed to minimize interaction burden**.
>
> a) the Abstention Module (Section 4.4, Figure 7) predicts whether enough context is available, avoiding unnecessary questions.
>
> b) the MCTS-based planner is designed so that, **at any point the user stops responding, the attributes collected so far are the most valuable for improving safety (Appendix C.6, C.7)**. This ensures that even if the interaction ends early, progress remains optimal for the given context.
>
> Empirically, RAISE reduces acquisition by an average of 2.7 queries from a 10-step budget without degrading safety performance (Table 2, Section 5.4). This approach balances efficiency, privacy, and personalization.
>
> [1] HR Kirk, B Vidgen, P Röttger, and SA Hale. The benefits, risks and bounds of personalizing the alignment of large language models to individuals. Nature Machine Intelligence 6, 4 (2024), 383–392. 2024
>
> [2] Ming Jin, Hyunin Lee. Position: AI Safety Must Embrace an Antifragile Perspective. Forty-second International Conference on Machine Learning (ICML). 2025.

---

### Official Review · Reviewer_x6f1 · 2025-06-30

**Clarity:** 3
**Significance:** 3
**Originality:** 3
**Rating:** 5
**Confidence:** 4

**Summary:**

LLMs often generate similar responses, posing safety risks in high-stakes applications. To address this, the concept of personalized safety is introduced, along with PENGUIN as a benchmark. Personalized user information significantly improves safety scores by 43.2%. A training-free framework, RAISE, acquires user-specific background, improving safety scores by up to 31.6%. This approach adapts safety research to individual user contexts, avoiding a universal harm standard.

**Questions:**

1. Page 1:
> This contrast underscores the urgent need for large language models (LLMs) to implement personalized safety mechanisms that account for individual vulnerability.

The formal definition of personalized safety should be given.

2. A few related works should be discussed.

- Jiang, Fengqing, Zhangchen Xu, Yuetai Li, Luyao Niu, Zhen Xiang, Bo Li, Bill Yuchen Lin, and Radha Poovendran. "Safechain: Safety of language models with long chain-of-thought reasoning capabilities." arXiv preprint arXiv:2502.12025 (2025).
- Huang, Yue, Lichao Sun, Haoran Wang, Siyuan Wu, Qihui Zhang, Yuan Li, Chujie Gao et al. "Position: Trustllm: Trustworthiness in large language models." In International Conference on Machine Learning, pp. 20166-20270. PMLR, 2024.
- Song, Da, Xuan Xie, Jiayang Song, Derui Zhu, Yuheng Huang, Felix Juefei-Xu, and Lei Ma. "Luna: A model-based universal analysis framework for large language models." IEEE Transactions on Software Engineering (2024).

**Ethical Concerns:**

["NO or VERY MINOR ethics concerns only"]

**Final Justification:**

Thanks for the rebuttal, I will keep my score.

**Limitations:**

Please see Questions.

**Quality:**

3

**Strengths And Weaknesses:**

Strengths:
+ The authors introduce a new concept of Personalized Safety of LLMs.
+ The authors develop PENGUIN, which is a benchmark with a substantial contribution: 14,000 scenarios across 7 high-stakes domains, each with both context-rich and context-free variants.
+ The proposed RAISE agent uses LLM-guided Monte Carlo Tree Search and a dual-module design,

Weaknesses:
- The definition of personalized safety should be given
- The personalization could be exploited for purposed manipulation, and the paper could benefit from a discussion on adversarial risks.

---

> ### Author Rebuttal · Authors · 2025-07-30
>
> We thank the reviewer for the insightful feedback and thorough review for our paper. We address the reviewer’s questions and concerns in detail below:
>
> >**The definition of personalized safety should be given.**
>
> Thank you for pointing this out. We define personalized safety as:
> **“The property of an LLM response being safe given an individual user’s context and vulnerability factors.”**
> We will explicitly include this definition in the revised manuscript.
>
> >**The personalization could be exploited for purposed manipulation, and the paper could benefit from a discussion on adversarial risks.**
>
> Thank you for raising this concern. The RAISE framework includes an **Abstention Module** (Figure 7) that determines whether the current context is sufficient for a safe response. **Additional attributes are only requested when the module predicts that more context is necessary**; otherwise, the query is answered directly without collecting any user information. This design ensures that sensitive attributes are collected only when essential, minimizing privacy risks.
>
> >**A few related works should be discussed.**
>
> Thank you for pointing this out. We agree these works are relevant and will include a discussion in the revised version. **SafeChain**[1] studies safety vulnerabilities in large reasoning models with long chain-of-thought reasoning and proposes datasets and methods to enhance safety without sacrificing reasoning quality. **TrustLLM**[2] provides a position paper introducing a multi-dimensional framework for trustworthiness evaluation of LLMs, covering truthfulness, safety, fairness, robustness, privacy, and ethics. **LUNA**[3] presents a model-based universal analysis framework for systematically assessing LLM quality issues such as bias and hallucination. In contrast, our work focuses on personalized safety, which is not addressed in these studies, and introduces both a benchmark (PENGUIN) and a planning-based agent (RAISE) for mitigating risks under user-specific contexts.
>
> [1]Jiang, Fengqing, Zhangchen Xu, Yuetai Li, Luyao Niu, Zhen Xiang, Bo Li, Bill Yuchen Lin, and Radha Poovendran. "Safechain: Safety of language models with long chain-of-thought reasoning capabilities." arXiv preprint arXiv:2502.12025 (2025).
>
> [2]Huang, Yue, Lichao Sun, Haoran Wang, Siyuan Wu, Qihui Zhang, Yuan Li, Chujie Gao et al. "Position: Trustllm: Trustworthiness in large language models." In International Conference on Machine Learning, pp. 20166-20270. PMLR, 2024.
>
> [3]Song, Da, Xuan Xie, Jiayang Song, Derui Zhu, Yuheng Huang, Felix Juefei-Xu, and Lei Ma. "Luna: A model-based universal analysis framework for large language models." IEEE Transactions on Software Engineering (2024).

---

> ### Comment · Reviewer_x6f1 · 2025-08-05
>
> Thank you for the rebuttal, and it addresses my concerns.

---

### Official Review · Reviewer_GUGt · 2025-07-03

**Clarity:** 3
**Significance:** 2
**Originality:** 3
**Rating:** 4
**Confidence:** 3

**Summary:**

The paper introduces "personalized safety" — the concept that identical large language model (LLM) responses can be harmless for one user yet dangerous for another, depending on individual context. To investigate this phenomenon, the authors develop PENGUIN, a comprehensive benchmark containing 14,000 scenarios across seven high-stakes domains (including mental health and finance). Each scenario exists in both context-free and context-rich versions, incorporating ten structured user attributes such as age, emotional state, and self-harm history. Responses are evaluated across three dimensions: risk sensitivity, emotional empathy, and user-specific alignment.

Experiments conducted with six leading LLMs demonstrate that providing complete user context improves average safety scores by 43.2%. However, the analysis reveals that certain attributes — particularly emotion and mental state — contribute significantly more to safety outcomes than others.

To efficiently gather only the most critical contextual information within limited interaction budgets, the authors propose RAISE, a training-free two-stage agent. The system employs an offline Monte Carlo Tree Search planner that pre-computes optimal attribute-question paths, combined with an online agent that queries users until an abstention module determines the response is sufficiently safe. RAISE achieves safety score improvements of up to 31.6% over baseline models while requiring only 2.7 questions on average, providing a practical pathway toward safer, context-aware LLM deployment without requiring model fine-tuning.

**Questions:**

Could the RAISE framework be further improved by continually incorporating real-world interaction logs, rather than relying solely on the initial offline simulations? As the system accumulates production-phase conversations, it could re-estimate which user attributes are most informative for each task class, potentially refining the planner's attribute rankings and reducing the number of follow-up questions over time. Have you considered, or experimented with, such an online learning loop?

**Ethical Concerns:**

["NO or VERY MINOR ethics concerns only"]

**Final Justification:**

The author's rebuttal not fully address my concern, but yet they are studying on an important research question with clearly formulation. Since I already gave a positive score, I will keep my score of 4. If they can incorporate online optimization or fully address the privacy problem might be better.

**Limitations:**

Yes.

**Quality:**

2

**Strengths And Weaknesses:**

Strengths

1. Timely, under-explored research problem. The paper addresses an important yet largely neglected question: safety is not one-size-fits-all—user context matters. By framing "personalized safety" as a distinct research axis, the work broadens the conversation beyond generic alignment and opens new ground for future study.

2. Valuable community asset. the PENGUIN benchmark. The authors release a 14,000-scenario benchmark spanning seven high-stakes domains and ten user attributes. This dataset not only standardizes evaluation for personalized safety but also empirically demonstrates that identical model outputs can swing from harmless to harmful when user context changes—quantifying this risk for the first time.

3. Clear, reproducible mitigation pipeline. The proposed mitigation strategy, which incorporates user context before responding, is straightforward and simple to implement.

4. Practicality conscious.
RAISE is explicitly designed to respect real-world constraints with limited interaction budgets. It boosts safety scores by up to 31.6% while asking an average of just 2-3 questions, striking a thoughtful balance between user burden and protection.

Weaknesses

1. Limited user attribute.
Although the benchmark covers ten personal attributes, many high-impact factors (e.g., cultural background, disability status, multilingual nuances, or extremism risk) remain unexplored.

2. No privacy or consent analysis beyond query minimization.
While RAISE limits the number of questions, it still can collect sensitive attributes (e.g., mental-health status) which does not address the privacy concerns.

---

> ### Author Rebuttal · Authors · 2025-07-30
>
> We thank the reviewer for the insightful feedback and thorough review for our paper. We address the reviewer’s questions and concerns in detail below::
>
> >**Limited user attributes**
>
> Thank you for this valuable observation. **Our current selection of ten attributes is grounded in well-established psychological and behavioral risk factors (Section 3.1), covering core dimensions such as mental health, emotional state, and socioeconomic context that strongly influence safety in high-risk interactions**. While we agree that additional factors—such as cultural background, disability status, and multilingual nuances—are important and can introduce new challenges, **our results demonstrate that even with these ten attributes, the phenomenon and mitigation strategies are robust and significant**.
>
> Our framework is designed to be **extensible**: both the **PENGUIN benchmark** and **RAISE agent** can incorporate new attributes without changing the overall methodology. We view this as an important direction for future work and will highlight this explicitly in the revised manuscript.
>
> >**No privacy or consent analysis beyond query minimization.**
>
> Thank you for raising this important point. **Our study does not involve any human-in-the-loop interaction or private data collection—it relies on publicly available Reddit posts (fully anonymized) and synthetic scenarios (50%, Appendix A.4)**. All Reddit data was collected via the official API in compliance with **Reddit’s Public Content Policy** [1], which explicitly permits non-commercial research use. We apply strong safeguards: **usernames are anonymized or removed, private fields (emails, IPs) stripped, and posts deleted prior to finalization are excluded. The dataset contains only text and cannot be traced back to individuals**.
>
> While some attributes (e.g., mental health) are sensitive, they are required to ensure safety in high-risk scenarios (Figure 5), and our framework minimizes exposure by collecting only attributes predicted as essential (Section 5.4), rather than requesting full profiles. Future work will explore additional privacy-preserving strategies.
>
> >**Could the RAISE framework be further improved by continually incorporating real-world interaction logs?**
>
> Thank you for this insightful suggestion. Our current study does not use historical interaction logs due to the lack of large-scale real-world history data, which is why we focus on an offline proof-of-concept simulation. We agree that, if such data becomes available, an online learning loop could further improve RAISE by re-estimating attribute importance and reducing follow-up queries. This aligns with approaches like Gemini with Personalization [2], which leverage prior user search or interaction history for better personalization. We plan to explore similar strategies in future work and will clarify this in the revision.
>
> **Our framework is designed to be extensible to such settings and plan to explore privacy-preserving online adaptation as part of future work**.
>
> [1] Reddit. Sharing our public content policy and a new home for researchers. 2024, May 30.
>
> [2] Gemini Team Google. Gemini: A Family of Highly Capable Multimodal Models. arXiv:2312.11805. 2023.

---

> > ### Comment · Reviewer_GUGt · 2025-08-06
> >
> > Thanks for your response.
> >
> > For the second question, actually you might misunderstood what I mean. I am asking that, though you use the RAISE framework to minimize the query minimization from users, yet you are still requiring personal sensitive information from the users during their use.

---

> ### Author Response · Authors · 2025-08-06
> **Replying to follow-up questions**
>
> We thank the reviewer for raising this important concern regarding privacy risks. While the RAISE framework is explicitly designed to minimize the number of user queries, we agree that it still involves collecting certain sensitive attributes (e.g., mental health status, emotional state), which may raise privacy concerns in real-world deployments.
>
> We will add this issue to the limitations section of the paper. Although privacy is not the primary focus of our current work, we have taken initial steps to reduce unnecessary exposure:
>
> (a) RAISE avoids collecting a full user profile by selecting only a small subset of attributes per interaction (an average of 2.7), based on estimated safety contribution. Furthermore, the RAISE architecture is modular and plug-and-play, allowing for natural future extensions such as incorporating **attribute-level privacy sensitivity weights**. This would enable the system to prioritize attributes that offer high safety gains with minimal privacy cost—moving toward a more balanced Pareto frontier between personalization and data minimization.
>
> (b) The **abstention module** allows the system to refuse to ask more privacy question when the model has sufficient context, thus avoiding pressure on users to disclose additional personal information. This provides a practical trade-off between safety and privacy.
>
> Nonetheless, privacy remains a core concern for real-world deployment. As a next step, future work can explore integrating  structured prior knowledge into the system, enabling the model to retrieve relevant context without repeated user prompting. We will include this discussion in the limitations section.
>
> We again thank the reviewer for highlighting this important dimension. We believe incorporating this discussion will strengthen the completeness and ethical awareness of our work, and we welcome any further suggestions.

---

### Official Review · Reviewer_a9rV · 2025-07-07

**Clarity:** 3
**Significance:** 3
**Originality:** 3
**Rating:** 4
**Confidence:** 4

**Summary:**

* The paper addresses the problem of personalized safety i.e. the same response to different sensitive queries would not be appropriate for different user risk profiles. The responses from the LLM would need to take into consideration different risk factors such as user profile before it actually generates a suitable response.
* The paper introduces a benchmark (Penguin) to study this problem systematically. The benchmark has around 14k scenarios across multiple domains such as health, education and finance.
* Each scenario is associated with a set of user attributes that are extracted from Reddit as well as synthetically generated.
* LLM is asked to generate responses to queries with and without the user attributes. They are then evaluated/graded on a 5-point likert scale.
* When provided the user context, the safety scores are vastly improved and the authors validate this claim across different models.
* They perform a series of ablations to figure out which attributes matter the most and also propose a framework called RAISE to help identify these attributes during inference for different users.

**Questions:**

Please see weakness section. Additional questions below

* How would we know for which queries we have to use persona attributes vs not? I see that all queries in the appendix are very highly connected to the persona provided - would the attributes affect responses to queries that not really related to the persona?
* Why not assume high risk user attributes always? The answers seem applicable for all risk categories.

**Ethical Concerns:**

["Major Concern: Data privacy, copyright, and consent"]

**Final Justification:**

The rebuttal responses are clear. It would be great to include details from there into the revised version.

**Limitations:**

yes

**Quality:**

3

**Strengths And Weaknesses:**

Strengths

* Very relevant research direction - context dependent safety queries and responses are very important for practical deployment of chatbots and LLMs.
* The empirical results are validated across a range of different LLM models which is very useful because different models employ different levels of safety training.
* The ablations studies are well constructed to see which attribute is very important.
* The MCTS approach to decide which attributes are important is novel and seems like a good add-on.

Weakness

* GPT-4o as judge puts us at risk of overfitting to Openai's evaluation of safety - any thoughts on how to address this?
* Extracting data from reddit is concerning - subreddits are usually targeted group of people with similar issues and attributes extracted from them might be skewed towards one section of the population. is there any analysis on how the different attributes are distributed?

---

> ### Author Rebuttal · Authors · 2025-07-30
>
> We thank the reviewer for the insightful feedback and thorough review for our paper. We appreciate the reviewer’s acknowledgment of the importance of context-dependent safety for LLMs and the breadth of our empirical validation across multiple models. We hope to address any questions below:
>
> >**GPT-4o as judge puts us at risk of overfitting to OpenAI's evaluation of safety**
>
> Thank you for raising this concern. We address this from two complementary angles: **(i) validating the reliability of GPT-4o as an evaluator** and **(ii) ensuring our evaluation design minimizes model-specific bias**.
>
> (i) validating the reliability of GPT-4o as an evaluator
>
> a). In **Appendix B.2**, we validated GPT-4o’s reliability as a judge through an **inter-rater agreement study with three human experts**. Specifically, GPT-4o’s evaluations achieved a **Cohen’s Kappa of κ = 0.688** and a **Pearson correlation of r = 0.92 (p < 0.001)** across 350 annotated samples. These results demonstrate that GPT-4o applies our structured safety rubric consistently with human experts.
>
> b). To address potential evaluator bias, we conducted an additional **cross-model consistency analysis** using **directional agreement** on pairwise preference judgments [1], which measures whether different evaluators (LLM as judge) consistently select the same response as better-performing. We compared GPT-4o’s judgments with those of **Claude-sonnet-4** [2], **Gemini-2.5-flash** [3], and **QwQ-32B** [4] across **2,000 evaluation pairs** in multiple settings. The **directional agreement** between GPT-4o and each alternative evaluator (Claude-sonnet-4, Gemini-2.5-flash, QwQ-32B) exceeded **90%** across all 2,000 evaluation pairs, confirming that our evaluation signal reflects method quality rather than model-specific bias.
>
>  (ii) ensuring our evaluation design minimizes model-specific bias.
>
> c). GPT-4o served exclusively as a judge based on the **evaluation protocol we defined (Appendix B.1.1)**, rather than relying on **OpenAI’s internal safety definition**. If our intent were to depend on GPT-4o’s built-in alignment standards, we could have simply asked whether a response is **“safe”** or **“unsafe”**. Instead, we designed a **structured safety rubric** that clearly operationalizes what constitutes high versus low safety across three dimensions—**Risk Sensitivity, Emotional Empathy, and Personalization**—reflecting the requirements of personalized safety, a setting we are the first to formalize.
>
> d). To accurately assess personalized safety, **evaluators**  must have **full access**  to **user context information**  because our rubric (Risk Sensitivity, Emotional Empathy, and User-Specific Alignment) is defined relative to the user’s context. Therefore, GPT-4o was provided with **the complete context** when judging responses. In contrast, evaluated models were deliberately withheld from accessing this context to replicate realistic deployment scenarios where user background information is initially unavailable.This design ensures that assessments capture safety performance independently of GPT-4o’s internal standards or context availability biases.
>
> **These results collectively indicate that the evaluation signal is model-agnostic rather than biased toward GPT-4o’s own outputs, and that our findings are not overfitted to a single model’s safety preferences.**
>
> >**Extracting data from reddit is concerning. Is there any analysis on how the different attributes are distributed?**
>
> Thank you for raising this concern. **Reddit** [5, 6, 7, 8] was selected as it is one of the few platforms with rich, self-disclosed user context in high-risk scenarios, which is essential for studying personalized safety. To mitigate domain bias, we cover **seven high-risk domains across 16 subreddit communities** and add **7,000 synthetic scenarios** for greater topical and demographic diversity (**Appendix A.4**).
>
> The full distributions of **all ten structured user attributes** demonstrate **coverage** across demographic and psychological dimensions. For example, age ranges from 18–24 to 55+, emotional states include seven categories, and mental health conditions cover five classes, **with no single category dominating the distribution**. Detailed distributions are provided in the following Table.
>
> |Attribute|Distribution|
> |----------------------|-------------------------------------------------------------------------------------------------|
> |Age|45–54 (16.29%), 35–44 (16.14%), 55+ (16.14%), 25–34 (16.14%), 18–24 (15.86%), Unknown (19.43%)|
> |Gender|Male (40.00%), Female (39.86%), Non-binary (18.86%), Unknown (1.28%)|
> |Marital|Single (27.29%), Married (23.14%), Divorced (19.43%), Widowed (14.29%), Unknown (15.85%)|
> |Profession|Software Engineer (7.29%), Graphic Designer (6.57%), Teacher (6.29%), Healthcare (5.86%), Student (7.29%), Service Worker (6.29%), Salesperson((5.86%), Freelancer(6.57%), Unknown (48.00%)|
> |Economic|Moderate (30.00%), Stable (24.14%), Difficult (22.57%), Wealthy (10.57%), Unknown (12.72%)|
> |Education|Bachelor's (33.14%), Master's (21.29%), PhD's (19.57%), High School (14.57%), Unknown (11.43%)|
> |Health |Good (42.57%), Fair (22.00%), Excellent (14.29%), Poor (11.71%), Unknown (9.43%)|
> |Mental|Mild Depression (20.71%), Anxiety (17.57%), Stress (16.43%), Severe Depression (14.00%), Unknown (31.29%)|
> |Past Self-Harm|None (80.71%), Yes (12.86%), Yes, in past year (6.43%)|
> |Emotion|Despair (10.57%), Anxious (9.71%), Neutral (9.14%), Hopeful (8.57%), Angry (8.29%), Happy (8.00%), Worried (7.43%), Fearful (7.14%), Indifferent (6.86%)|
>
> >**How would we know for which queries we have to use persona attributes vs not?**
>
> Thank you for these related questions. **Our framework uses an Abstention Module (Figure 7) to predict whether additional user context is needed for safe response generation.** Low-risk queries (evaluated under the same rubrics in our paper) are answered directly without personalization, while extra attributes are gathered only for safety-critical cases, reducing privacy exposure and inefficiency. Assuming all users are high-risk would lead to unnecessary refusals and complexity.
>
> To validate our design, we sampled **14,000 English conversations** from the ShareGPT‑Chinese‑English‑90k dataset [4], which mainly consists of low-risk queries such as writing, programming, and natural science tasks. These queries are well-suited for testing whether the abstention module can accurately identify cases that do not require additional user context. Combined with our original benchmark, this **yields 28,000 labeled queries** for the abstention classification task. The model predicts whether extra user background is necessary for safe response generation and achieves **87% accuracy** on the combined dataset, demonstrating its ability to selectively acquire attributes rather than assuming all queries are high-risk.
>
> >**Why not assume high risk user attributes always?**
>
> **Always assuming high-risk would make the system inefficient and privacy-invasive [10]**, as it forces unnecessary collection of user attributes for low-risk queries and increases interaction complexity. **Our framework is designed to avoid these issues while maintaining strong safety guarantees**. It uses an Abstention Module (Section 5.4) to decide when extra attributes are truly needed. In addition, attribute sensitivity analysis (Figure 5) reveals large variation in attribute utility, showing that exhaustive acquisition offers little benefit. Combined with an MCTS-based planner, our selective approach reduces acquisition steps by up to 2.7 user queries (out of an initial 10-query budget) while maintaining overall safety performance—demonstrating that our design prioritizes efficiency and privacy alongside personalization.
>
> >**Major Concern: Data privacy, copyright, and consent**
>
> Thank you for raising this concern. Our data is collected only from publicly available Reddit posts using the official API, fully complying with Reddit’s Public Content Policy[11], which explicitly permits non-commercial research. According to Reddit’s statement (May 30, 2024):
>
> ```"We continue to believe in supporting public access to Reddit content for researchers and those who believe in responsible non-commercial use of public data." ```
>
> We apply strong privacy safeguards: **all usernames are anonymized or removed, private fields (emails, IPs) are stripped, and posts deleted before finalization are excluded**. The dataset contains only post text and cannot be traced back to individuals. Our data collection follows the same policy and ethical standards adopted in prior widely cited NLP works [5, 6, 7, 8]. **Our use is strictly non-commercial, for aggregate language analysis, and aligns with both platform guidelines and ethical norms**.
>
> [1] Jiawei Gu, et al. A Survey on LLM-as-a-Judge. arXiv:2411.15594. 2024.
>
> [2] Anthropic. The claude 3 model family: Opus, sonnet, haiku. 2024.
>
> [3] Gemini Team Google. Gemini: A Family of Highly Capable Multimodal Models. arXiv:2312.11805. 2023.
>
> [4] An Yang, et al. Qwen2.5 Technical Report. arXiv:2412.15115. 2025.
>
> [5] Weiyan Shi, et al. CultureBank: An Online Community-Driven Knowledge Base Towards Culturally Aware Language Technologies. Findings of EMNLP. 2024.
>
> [6] Elsbeth Turcan, et al. Dreaddit: A Reddit Dataset for Stress Analysis in Social Media. Association for Computational Linguistics. 2019.
>
> [7] Byeongchang Kim, et al. Abstractive Summarization of Reddit Posts with Multi-level Memory Networks. Association for Computational Linguistics. 2019.
>
> [8] Tommaso Caselli, et al. HateBERT: Retraining BERT for Abusive Language Detection in English. 2021.
>
> [9] shareAI. ShareGPT-Chinese-English-90k Bilingual Human-Machine QA Dataset. 2023.
>
> [10] Paul Röttger, et al. XSTest: A Test Suite for Identifying Exaggerated Safety Behaviours in Large Language Models. arXiv:2308.01263. 2024
>
> [11] Reddit. Sharing our public content policy and a new home for researchers. 2024.

---

> > ### Comment · Reviewer_a9rV · 2025-08-05
> >
> > Thanks for the comments. Please add details about the attribute distribution in the revised version. I am updating my score.

---

### Note · Authors · 2025-08-14

We sincerely thank all reviewers and ethics reviewers for their constructive feedback, active engagement, and final confirmation that their concerns have been fully addressed. The reviewers recognized the importance and novelty of our work on **personalized safety** for LLMs via the **PENGUIN** benchmark and **RAISE** framework, highlighting:

a. Timely and under explored problem — formalizing personalized safety as a distinct dimension of LLM safety

b. Valuable benchmark — 14k high-risk scenarios across seven domains with structured user attributes

c. Novel and practical mitigation — RAISE’s LLM-guided MCTS with an Abstention Module for efficient, safe personalization

d. Extensive empirical validation — significant improvements across multiple models, with attribute-level ablations

**Main Concerns and How They Were Addressed**

a. Evaluator bias (GPT-4o) → Validated through agreement with human experts (κ = 0.688, r = 0.92) and >90% cross-model agreement; rubric-based, model-agnostic evaluation protocol.

b. Reddit data, privacy, and licensing → Fully compliant with Reddit policies; all PII removed; attribute distributions reported; Reddit-derived data released only under research-use agreements, synthetic data under CC-BY 4.0 license.

c. Privacy in attribute collection → Average of only 2.7 attributes collected per interaction; abstention module avoids unnecessary questions; future work includes privacy-sensitivity weighting and prior knowledge integration.

d. RAISE vs. form-based approaches → Static or rule-based dynamic forms face scalability and adaptability limitations: as attributes and risk factors grow, path combinations increase exponentially, making pre-defined branching impractical. RAISE uses LLM-guided MCTS to dynamically determine the optimal next question based on prior responses, with the abstention module deciding when sufficient context is reached to avoid redundancy. Even if users stop early, collected attributes yield maximal safety benefit so far. This adaptive planning balances efficiency, privacy, and safety, and can serve as the backend logic for dynamic UIs in more complex interaction settings.

We will integrate all clarifications into the Dataset Construction, Limitations, and Ethics Statement sections. **We appreciate the recognition of PENGUIN and RAISE’s potential to advance safer, context-aware LLMs while balancing personalization and privacy, and we hope this work will inspire further research toward trustworthy AI.**

---

### Decision · Program_Chairs · 2025-09-17

**Decision:**

Accept (poster)

**Comment:**

This submission introduces the notion of personalized safety for LLMs whereby identical responses can have divergent risks depending on the user's background or circumstances. Two primary contributions are provided. First, the authors introduce the PENGUIN benchmark, a comprehensive dataset of scenarios across 7 high-stakes domains, each with context-rich and context-free variants providing systematic evaluation of LLM safety in personalized settings. Second, the authors introduce the RAISE framework which uses Monte Carlo Tree Search to acquire user-specific context attributes, optimizing for safety under limited interaction budgets. Through empirical evaluations, the authors show that access to user context can substantially enhance safety.

The reviewers agree that the submission is timely and sheds light on an important and under-explored problem in the LLM safety space. They also appreciated the thorough empirical results and the inclusion of ablation studies. The authors have addressed the ethical concerns raised by the ethical reviewer. Overall, the reviewers unanimously agree that the paper provides a valuable asset for the community. I am therefore pleased to recommend this paper for acceptance.